# Comparison of Four Treatment Protocols with Intra-Articular Medium Molecular Weight Hyaluronic Acid in Induced Temporomandibular Osteoarthritis: An Experimental Study

**DOI:** 10.3390/ijms241814130

**Published:** 2023-09-15

**Authors:** Schilin Wen, Veronica Iturriaga, Bélgica Vásquez, Mariano del Sol

**Affiliations:** 1Doctoral Program in Morphological Sciences, Faculty of Medicine, Universidad de La Frontera, Temuco 4780000, Chile; schilin.wen@cloud.uautonoma.cl; 2Grupo de Investigación de Pregrado en Odontología, Facultad de Ciencias de la Salud (FACSA), Universidad Autónoma de Chile, Temuco 4810101, Chile; 3Temporomandibular Disorder and Orofacial Pain Program, Department of Integral Adult Care Dentistry, Universidad de La Frontera, Temuco 4780000, Chile; veronica.iturriaga@ufrontera.cl; 4Sleep & Pain Research Group, Faculty of Dentistry, Universidad de La Frontera, Temuco 4780000, Chile; 5Center of Excellence in Morphological and Surgical Studies (CEMyQ), Universidad de La Frontera, Temuco 4780000, Chile; 6Department of Basic Sciences, Faculty of Medicine, Universidad de La Frontera, Temuco 4780000, Chile; belgica.vasquez@ufrontera.cl

**Keywords:** temporomandibular joint, osteoarthritis, hyaluronic acid, molecular weight, viscosity

## Abstract

The aim was to compare the effect between a single intra-articular infiltration (1i) and two infiltrations (2i) of medium molecular weight hyaluronic acid (MMW-HA) of high viscosity (HV) and low viscosity (LV) on the histopathological characteristics of temporomandibular joint (TMJ) osteoarthritis (OA) induced in rabbits. An experimental study was conducted on *Oryctolagus cuniculus* rabbits, including 42 TMJs, distributed between (1) TMJ-C, control group; (2) TMJ-OA, group with OA; (3) TMJ-OA-wt, group with untreated OA; (4) group treated with HA-HV-1i; (5) group treated with HA-HV-2i; (6) group treated with HA-LV-1i; and (7) group treated with HA-LV-2i. The results were evaluated using the Osteoarthritis Research Society International (OARSI) scale and descriptive histology considering the mandibular condyle (MC), the articular disc (AD), and the mandibular fossa (MF). The Kruskal–Wallis test was used for the statistical analysis, considering *p* < 0.05 significant. All treated groups significantly decreased the severity of OA compared to the TMJ-OA-wt group. The HA-HV-2i group showed significant differences in the degree of OA from the TMJ-OA group. The degree of OA in the HA-HV-2i group was significantly lower than in the HA-LV-1i, HA-LV-2i, and HA-HV-1i groups. The protocol that showed better results in repairing the joint was HA-HV-2i. There are histological differences depending on the protocol of the preparation used: two infiltrations seem to be better than one, and when applying two doses, high viscosity shows better results.

## 1. Introduction

### 1.1. Background

The articular cartilage of the temporomandibular joint (TMJ) is composed mainly of chondrocytes and an extracellular matrix (ECM) that contains type I and II collagen fibers and proteoglycans, which give it structure and resistance. The cartilage presents three clearly identifiable zones: the superficial zone (SZ), middle zone (MZ), and deep zone (DZ). The SZ is characterized by the presence of chondrocytes arranged in rows parallel to the articular surface, the MZ shows a gradual transition between the chondrocytes aligned in rows and the chondrocytes arranged in larger groups, and the DZ has larger chondrocytes arranged in groups, with a denser ECM. The subchondral trabecular bone (SB) contains blood elements in its bone marrow. The articular disc (AD) comprises fibrocartilage, the chondrocytes of which follow the order of collagen fibers [1]. Osteoarthritis (OA) is a chronic degenerative disease affecting different TMJ components, including the articular cartilage [2]. In OA, the cartilage can suffer erosion, wear and tear, thinning, formation of cracks and fibrillations, and loss of structure and normal organization [1,3]. These changes in the cartilage can increase the friction between the bone surfaces, cause inflammation, and reduce the ability to absorb impact during mandibular movements, which can result in pain, stiffness, and limitation of the function of the mandible [4].

On the other hand, hyaluronic acid (HA) is a glycosaminoglycan polysaccharide secreted by type B synoviocytes and is produced naturally in cartilage and synovial fluid [5].

Intra-articular injections of exogenous HA into the TMJ allow viscosupplementation, that is, the administration of the main component of the synovial fluid [6,7]. This treatment continues to be the minimally invasive treatment of choice due to its satisfactory results in terms of pain relief and improvement of mandibular function, and due to its low incidence of side effects [8,9]. Moreover, additional biological effects have been observed, including reduced inflammation, increased lubrication, decreased levels of proinflammatory mediators, and the promotion of articular cartilage repair [10,11].

According to their molecular weight (MW), the exogenous HA preparations can be categorized as low molecular weight (LMW-HA) (0.5–1 × 10^6^ Da), medium molecular weight (MMW-HA) (1.2–4.5 × 10^6^ Da), and high molecular weight (HMW-HA) (6–7 × 10^6^ Da) [12]. It is proposed that the properties of exogenous HA, like its MW, viscosity, and infiltration frequency, have various effects on TMJ-OA. In in vitro studies using samples of synovial tissue and osteoarthritic knee cartilage, it has been noted that HMW-HA infiltration significantly inhibits the activity of metalloproteinases (MMPs) compared to LMW-HA [13]. However, it has been described that the chondroprotective effect of HA remains, regardless of its MW [14]. In a study by Duygu [15] on rabbits with induced TMJ-OA, the histological effects of the infiltration of HMW-HA and physiological serum were compared. It was observed that the HMW-HA showed a significantly more effective improvement in joint repair at four weeks post-treatment, although at 6 and 8 weeks, no differences were observed between the two groups.

Tang [16] described MMW-HA (1.5–2.5 × 10^6^ Da) as being more effective in the treatment of TMJ-OA than a physiological saline solution, significantly reducing the activity of the enzymes and receptors involved in the activation of the MMPs responsible for degrading the ECM of the cartilage during the pathogeny of TMJ-OA. Iturriaga [1] compared the effects of HMW-HA and LMW-HA in treating TMJ-OA induced in rabbits. An improvement was noted in both groups compared with the untreated group. However, LMW-HA showed a significantly greater improvement in the cartilage and the AD. In terms of the viscosity of exogenous HA, there remains a lack of clarity concerning whether different viscosities have different clinical and histopathological effects. In in vitro conditions, it has been observed that all the variants of exogenous HA viscosity show a thinning behavior due to viscoelastic shear, which implies that they have a lubricant effect on the articular cartilage [17].

In terms of the frequency of the intra-articular infiltrations of HA, it has been reported clinically that both single and multiple infiltrations have been effective in pain relief for patients with TMJ-OA, both at rest and during mastication, in addition to improving the mouth opening range [11,18,19].

### 1.2. Rationale

To date, the clinical effectiveness of exogenous HA has been demonstrated [20,21,22]. Furthermore, the rheological properties (such as molecular weight, concentration, and viscoelasticity) of exogenous HA are important determining factors for the success of OA treatment. It is also reported that the degree of anti-inflammatory, immunomodulatory, and analgesic effects of HA is determined by the MW of the HA [23].

However, due to the heterogeneity of the studies, it has not been possible to establish the effectiveness of HA in OA, with multiple protocols described in relation to molecular weight, concentration, viscosity, and number of infiltrations.

### 1.3. Aim

Therefore, this study aimed to compare the histopathological effects between the single intra-articular infiltration and the two-time infiltration of MMW-HA of high viscosity (HV) and low viscosity (LV) in a model of TMJ-OA induced in rabbits.

## 2. Results

### 2.1. TMJ-C

The articular cartilage of the mandibular condyle (MC) and mandibular fossa (MF) was characterized as presenting three well-defined zones: SZ, MZ, and DZ. The surface was smooth and continuous. The matrix, the organization of the collagen, and the size, shape, and distribution of the chondrocytes were characteristic of each zone. The DZ was the thickest, and the chondrocytes were round and larger. The tidemark was not defined. The calcified cartilage (CC) was located between the tidemark and the SB. The SB was spongy, with a predominance of blood elements in its bone marrow. The AD was inserted in the articular capsule between the MC and the MF. The disc was formed by avascular fibrous cartilage. The chondrocytes were arranged in rows relative to the collagen fibers (Figure 1A–D).

### 2.2. TMJ-OA

The articular cartilage of MC and MF was characterized by less thickness and by focal and irregular surface discontinuity. The matrix presented deep fibrillations that extended from the surface to the DZ. In the most severe cases, branched fibrillations were observed. The matrix texture became more heterogeneous, with adjacent domains of proteoglycan depletion and increased cationic staining being observed. The death of chondrocytes and disoriented chondrons and chondrocytes were evident, mainly in the chondrons close to the fibrillations. The AD presented disorganized collagen fibers, with hypertrophic chondrocytes arranged randomly (Figure 1E–H). The degree and stage of the OA in the MC and MF are observed in Table 1.

### 2.3. TMJ-OA-wt

The MC and MF articular cartilage presented greater progression in the degree of OA than the TMJ-OA group, although with no significant differences in the score on the OARSI scale (*p* = 0.063; *p* = 0.111, respectively) (Table 1). The AD presented characteristics similar to the TMJ-OA group. The hypertrophy of the synovial membrane as far as the articular cavity was evident. The histological characteristics of this group are shown in Figure 2A–D.

### 2.4. Groups Treated with MMW-HA

The histopathological analysis showed that all the groups treated with MMW-HA, regardless of the viscosity and the number of infiltrations, presented a decrease in the degree of OA in the articular cartilage of both the MC and the MF. In addition, the AD showed a reordering of collagen fibers, presenting a more parallel arrangement between them. Table 2 describes the histological characteristics of the TMJ-OA of the different treatment groups. Figure 2E–T shows the histological characteristics of the same groups.

The OARSI scale score showed that all the treated groups significantly reduced the severity of the osteoarthritic process of the cartilage of the MC and MF compared to the TMJ-OA-wt group (Table 1). The HA-HV-2i group showed significant differences in the degree of OA, in both the MC and MF, with the TMJ-OA group (*p* = 0.006; *p* = 0.04, respectively) and with the TMJ-OA-wt group (*p* = 0.008 *p* = 0.008, respectively). The HA-LV-2i group showed significant differences in the MC and MF, only with the TMJ-OA-wt group (*p* = 0.0001; *p* = 0.0001, respectively). None of the treated groups significantly reduced the horizontal extension of the cartilage involvement (Table 1).

When comparing the treated groups, the OARSI scale score showed that the degree of OA in the cartilage of the MC of the HA-HV-2i group was significantly smaller than the HA-LV-2i group (*p* = 0.04). Furthermore, it presented significant differences in the degree of OA in the MF compared to the HA-LV-1i and HA-HV-1i groups (*p* = 0.02; *p* = 0.04, respectively) (Table 1).

## 3. Discussion

It has been observed that the synovial fluid (SF) in the OA presented changes in its composition, viscoelasticity, and molecular weight (MW) in comparison with the normal SF [24,25,26]. Normal SF has an approximate MW of 2 to 3 × 10^6^ Da, whereas, in the OA, its MW is less than 0.6 × 10^6^ Da [27,28]. The normal endogenous HA, a key component of SF, also undergoes a reduction in its MW due to degradation by free radicals and rupture of its chains in OA [29,30]. This degradation of HA contributes to a reduction in the viscoelasticity of the SF and its ability to protect the joint [31,32].

The intra-articular infiltration of exogenous HA is one of the main treatments in TMJ-OA. However, the rheological properties of the different formulations, like the friction coefficient, viscoelasticity, and shear velocity, vary widely from each other [17]. This variability is also reflected in clinical differences [33].

Viscosupplementation with exogenous HA is based on restoring the viscosity observed in the pathological SF after the intra-articular infiltration of HA [34]. This property represents the ability of the HA molecule to resist permanent deformation under a long-term load [35,36,37]. Although exogenous HA is detected in the SF only a few days after the injection, the positive clinical effects of the infiltration remain even after 6 months, which suggests exogenous HA also promotes the synthesis of endogenous HA [38,39]. 

The MW of exogenous HA is a determining factor in its biological activity [14,38]. It has been suggested that the ideal range of MW for the preparations of exogenous HA must be between 0.5 and 4.0 × 10^6^ Da to imitate endogenous HA. This facilitates the access of HA to the cells through an endocytic pathway and its interaction with specific intracellular proteins [40]. It has been observed that the intra-articular infiltration of MMW-HA in the range of 1.5 to 2 × 10^6^ Da improves the signs and symptoms of temporomandibular disorders [41], and, at the histological level, MMW-HA between 2.3 and 2.5 × 10^6^ Da has proven to be significantly more effective in the repair of the articular cartilage, as well as in the synovial membrane and the AD of rats with TMJ-OA, compared with a saline solution [42]. In addition, MMW-HA between 1.5 and 2.5 × 10^6^ Da significantly reduces the activity of the enzymes and receptors involved in the degradation of the cartilage matrix by MMPs [16]. In our study, TMJ-OA was treated in rabbits with MMW-HA in a range of 1.1 to 2.1 × 10^6^ Da, which resulted in a reduction of OA in the articular cartilage and articular disc, in all the protocols used, independent of the viscosity and number of infiltrations used. However, this reduction was statistically significant when the groups treated with HA of MMW and high viscosity were compared, where the application of two infiltrations significantly decreased the severity of OA compared to a single infiltration. When the groups treated with two infiltrations were compared, it was observed that those who received high-viscosity HA had significantly better joint repair. Therefore, the MMW-HA-HV-2i protocol had the best results. These findings are consistent with those obtained by Iturriaga [1], who used HMW-HA to treat TMJ-OA in rabbits. However, it was observed that the MMW-HA used in our study resulted in a higher score in the degree of OA according to the OARSI scale compared to the scores reported by Iturriaga [1] when using LMW-HA. In this regard, some of the reasons why LMW-HA may further decrease the degree of osteoarthritis of the articular cartilage may be due to its greater ability to penetrate the tissues and cells of the TMJ. This has potential benefits since a greater viscosity of HA is associated with better viscosupplementation and viscoelasticity. In addition, being both a MMW-HA and LMW-HA, it allows for enhanced steric interactions compared to HMW-HA preparations, thereby promoting viscoinduction.

This allows it to exert more direct and specific effects on the articular cartilage, promoting its regeneration and reducing inflammation effectively [43]. In addition, it may have a greater ability to modulate the biological processes involved in osteoarthritis, such as inflammation and the activity of the enzymes and receptors involved in the degradation of the articular cartilage [44]. This can help prevent the progression of the disease and promote the repair of the damaged cartilage. It is important to consider, however, that the results can vary in different studies and clinical cases. In this regard, the study by Guarda-Nardini [19] compared the effects of LMW-HA and MMW-HA in patients with TMJ-OA and found no statistically significant differences in terms of improvement of the osteoarthritis symptoms in the TMJ. On the other hand, it has been reported that exogenous HA therapy with a higher MW was more effective in relieving OA-associated pain than therapies with a lower MW [45]. Therefore, the choice of the type of hyaluronic acid to use must be based on an individualized assessment of each patient.

The therapeutic impact of the viscosity of exogenous HA on the treatment of TMJ osteoarthritis has not yet been conclusively determined. While some studies, like that of Rebenda [46], suggest HMW-HA with a greater viscosity can have greater benefits in terms of viscosupplementation and lubrication than with LMW-HA, previous studies by Bonnevie [17] and Machado [47] have posited that the viscosity of an HA viscosupplement is not necessarily related to its clinical effectiveness. Moreover, possible disadvantages associated with HMW-HA have been indicated, such as limited molecular mobility within the joint and a reduced viscoinductive effect on the production of endogenous hyaluronic acid and promoter molecules of the joint repair [1,48,49]. Generally, to achieve greater viscosity in exogenous HA, its molecular weight is increased. However, in the TMJ, it appears that intra-articular infiltration of HMW-HA would present a lower repair capacity of articular fibrocartilage compared to medium and low MW HA, due to the reasons discussed above. In this sense, the use of MMW-HA could be a good alternative for the clinician, presenting a balance between viscosity and MW.

On the other hand, the number of intra-articular infiltrations in OA treatment has also been controversial due to the diversity of protocols described in the literature. Iturriaga [1] conducted a study on rabbits to assess the effect of a single infiltration of exogenous HA (HMW and LMW) in managing TMJ-OA. A repair of the MC and the MF cartilage was observed 30 days after the hyaluronic acid infiltration in both groups. However, 135 days after the infiltration, both groups showed a deterioration in the joint repair, with more fibrillations in the cartilage and hypertrophy of the synovial membrane, which suggests the possible need to repeat the infiltration over time [1].

In clinical trials, a reduction in pain and an improvement in mandibular function have been observed in both single and multiple infiltrations in the short term [15,18,19]. However, it has also been described that the intensity of pain 6 months after the start of treatment is positively correlated with the number of injections per joint, the total amount of drug administered in milliliters, and the volume of drug administered monthly per joint. These correlations are not strong, but they are enough to raise suspicions about whether repeated interventions reduce the analgesic effect; that is, fewer administrations of HA may be more effective in controlling TMJ pain [50].

In clinical trials that have used protocols of multiple injections of exogenous HA as a treatment for TMJ-OA, a gradual reduction in pain and a progressive improvement in masticatory function have been noted over time, demonstrating the effectiveness of repeated injections [51]. These results are consistent with those obtained in other joints in the body [52,53]. In this study, it was evident that when two doses of HA were given, the scores on the OARSI scale were lower than when only one dose was applied, in both HA-LV and HA-HV. HA-HV applied in two doses was the protocol that had the best results.

Although there is clinical evidence of the advantages and effects of HA in the treatment of temporomandibular disorders [54], histological evidence is still scarce. In recent years, studies have been published that evaluate the effects of HA by varying its rheological properties [1,33], and significant differences have been observed in cartilage repair, as well as in the inflammatory response [55,56].

We highlight the value of this study, where an antiarthritic effect of MMW-HA is evident in all the protocols evaluated, allowing the viscosity of the solution to be varied without losing the reparative effect of the articular cartilage. It is also important to consider the repetition of the application of exogenous HA over time, considering the better results when applying two doses than a single dose. Regarding this, we must not forget that the application interval between the first and second dose was 15 days in this study; however, animal bone metabolism is not the same as that of humans, making it necessary to consider the differences in the times between the results in animal models and studies in humans, where these 15 days probably correspond to a few months in humans.

It is suggested to conduct more clinical studies with a large sample size that compare protocols of different viscosity and infiltration frequency in patients with TMJ-OA, evaluating pain, mandibular range of motion, and joint sounds.

### Limitations

This study presents some limitations that must be taken into consideration. First, it is important to recognize the difficulty of extrapolating the results obtained in the animal model to the human reality. Although the findings provide valuable information, the anatomical and physiological differences between species can affect the results and their direct clinical applicability.

Moreover, the lack of long-term follow-up with and without new infiltrations in this study must be mentioned. The longer-term evaluation would make it possible to better understand the persistence of the therapeutic effects and determine if additional infiltrations are required to maintain the long-term benefits. Consequently, additional studies on humans are required that address these limitations and provide a more complete view of the therapeutic effects of HA infiltrations on osteoarthritis in the TMJ.

## 4. Materials and Methods

An experimental study was conducted on 21 *Oryctolagus cuniculus* rabbits, males, healthy, weighing 3 kg, and 8 months old, according to the recommendations described by Poole [57]. The animals were maintained in an environment controlled for temperature, environmental noise, and a cycle of 12 h light/12 h dark. The animals were housed in cages individually and randomly. The animals were in the care of a veterinarian, following the ARRIVE guidelines and the National Research Council’s Guide for the Care and Use of Laboratory Animals [58].

The study was carried out in the experimental surgery unit of the Center of Excellence in Morphological and Surgical Studies at the Universidad de La Frontera, Chile, with the approval of the Universidad de La Frontera’s Scientific Ethics Committee (File Number 090_21). Both TMJs were considered for each animal to reduce the sample size and comply with the experimentation criteria established by Russell and Burch [59]. Groups were randomly assigned, and the allocation sequence was hidden. Forty-two TMJs were included, distributed by (1) TMJ-C, a control group comprising 6 healthy TMJs; (2) TMJ-OA, a group comprising 6 TMJs with OA; (3) TMJ-OA-wt., a group comprising 6 untreated TMJ-OA; (4) HA-HV-1i, a group comprising 6 TMJ-OA treated with MMW-HA of HV with a single infiltration; (5) HA-HV-2i, a group comprising 6 TMJ-OA treated with MMW-HA of HV with two infiltrations two weeks apart; (6) HA-LV-1i, a group comprising 6 TMJ-OA treated with MMW-HA of LV with a single infiltration; and (7) HA-LV-2i, a group comprising 6 TMJ-OA treated with MMW-HA of LV with two infiltrations two weeks apart. The group distribution is shown in Figure 3. The animals were housed individually and randomly in cages when the procedures were complete.

The protocols for the TMJ-OA induction, intra-articular HA infiltration, histological processing, and histological analysis were performed according to previously described protocols [1,15,60,61,62].

### 4.1. TMJ-OA Induction

TMJ-OA was induced in all the animals except in the TMJ-C group. The animals were anesthetized intramuscularly with ketamine (40 mg/kg), xylazine (5 mg/kg), and acepromazine (1 mg/kg) using the technique described by Iturriaga [1,60]. OA was induced by infiltration of sodium monoiodoacetate (MIA) at a concentration of 3 mg/mL in the articular space with a 22G caliber needle. It took a 50-day waiting period to develop TMJ-OA [1,15]. After the 50-day waiting period, the animals in the TMJ-OA group were euthanized and then analyzed. In the TMJ-OA-wt, HA-HV-1i, HA-HV-2i, HA-LV-1i, and HA-LV-2i groups, after the initial 50-day period, 30 additional days were taken prior to euthanasia and analysis (Figure 3).

### 4.2. Intra-Articular Infiltration of HA

The intra-articular infiltration of HA was conducted using the same protocol for the anesthesia and previously mentioned preoperative measures. The infiltration technique was standardized, considering the caudal margin of the orbital lamina as an anatomical reference point, and was directed 5 mm caudally and 1 mm ventrally, with the needle at a 45° angle ventral to the skin [60]. A preparation of MMW-HA of HV and MMW-HA of LV was used. In the HA-HV-1i and HA-HV-2i groups, 0.1 mL of exogenous HA of MMW and HV was applied (Sysons 1.6, Cosmepharma, Chile: concentration of 1.6 mg/mL; molecular weight of 1.1 to 2.1 × 10^6^ Da; viscosity of 80 mPA). In the HA-LV-1i and HA-LV-2i groups, 0.1 mL of HA of MMW and LV was applied (Sysons 1.0 Cosmepharma, Chile: concentration of 1.0 mg/mL; molecular weight of 1.1 to 2.1 × 10^6^ Da; viscosity of 30 mPA).

### 4.3. Histological Processing

After the euthanasia of the animal, the articular tissue was dissected. There were animal losses, adverse effects, or modifications. The samples were coded to maintain the masking of the process. The tissue was fixed with 10% buffered formalin (formaldehyde 1.27 mol/L in phosphate buffer of 0.1 M, pH 7.2) for 48 h. Then, the samples were decalcified in ethylenediaminetetraacetic acid (EDTA) (in 0.1 M phosphate buffer 7–8) [61], in ultrasonic decalcification (Use 33, Medite, Burgdorf, Germany) for 30 days. Next, the samples were dehydrated in ascending alcohol, clarified with xylol, and embedded in Paraplast Plus (Sigma-Aldrich Co., St. Louis, MO, USA). Serial sections of the TMJ were cut in the parasagittal plane at a thickness of 5 μm using a Leica rm2255 microtome. To optimally score the OA, successive sections of the deepest planes of the joint were stained and viewed under a light microscope. Then, for the more detailed analysis, a single section per joint was selected, considering the plane of the block that crosses the lesion to the greatest extent and that presents the most pronounced alterations [62]. The histological sections were stained with Toluidine blue stain, visualized under an optical microscope (Leica DM 2000 LED, Wetzlar, Germany), and photographed with a digital camera (Leica high-definition MC 170, Wetzlar, Germany). The slides and photographs were also coded to maintain the masking during this process.

### 4.4. Histological Analysis

A descriptive analysis of the MC, AD, and MF was performed. The cartilage was described from superficial to deep, beginning with the SZ, followed by the MZ, the DZ, the CC, and the SB. In reference to the AD, the central zone was analyzed at its thinnest point, as well as the anterior and posterior peripheral areas. For the histopathological evaluation of the OA in the MC and the MF, the OARSI scale was used, considering the description by Pritzker [62] and later recommendations [63,64]. The OARSI scale evaluated the degree of damage to the articular cartilage and the stage corresponding to the extent of the damage. The degree of damage is classified from 0 to 6 degrees, where degree 0 is normal tissue and degrees 1 to 6 are related to OA. Degrees 1 to 4 of OA imply changes only to the cartilage, whereas degrees 5 and 6 also include the SB. In relation to the stages, the OARSI scale defines 4 stages based on the horizontal extension of the affected cartilaginous surface, regardless of the degree of the underlying OA. Stage 1 represents involvement below 10%; stage 2 involvement of 10–25%; stage 3 involvement of 25–50%; and stage 4 involvement greater than 50%.

The histological analysis was conducted independently by two authors (SW and BV). Disagreements were discussed until a consensus was reached. The interobserver calibration was conducted in three phases before analyzing the samples: theoretical training, laboratory training, and the calibration itself (final kappa agreement 0.79).

### 4.5. Statistical Analysis

The data analysis is shown in the median and interquartile range. Non-parametric inferential statistics were performed comparing both the degree and stage of OA between the intervention groups, using the Kruskal–Wallis test and post hoc Dunn’s test. The analysis was performed with the STATA 18 program, considering a significance level of α = 0.05.

## 5. Conclusions

The results of this study show that the use of exogenous MMW-HA has a positive effect on the treatment of TMJ-OA. All the groups treated with MMW-HA showed a reduction in the severity of TMJ-OA in the articular cartilage of the MC, AD, and MF, with statistically significant differences compared to the TMJ-OA-wt and TMJ-OA groups.

The protocol of two infiltrations of HA of high viscosity (HA-HV-2i) showed an additional improvement in the repair of the articular cartilage in both the MC and MF. A significant reduction in deep fibrillations, an increase in the number of cells present, and a reorganization of the chondrocytes were observed in the group that received HA-HV-2i. These results highlight the potential benefits of using the protocol of two infiltrations of high-viscosity HA to improve the health of the articular cartilage in TMJ-OA.

## Figures and Tables

**Figure 1 ijms-24-14130-f001:**
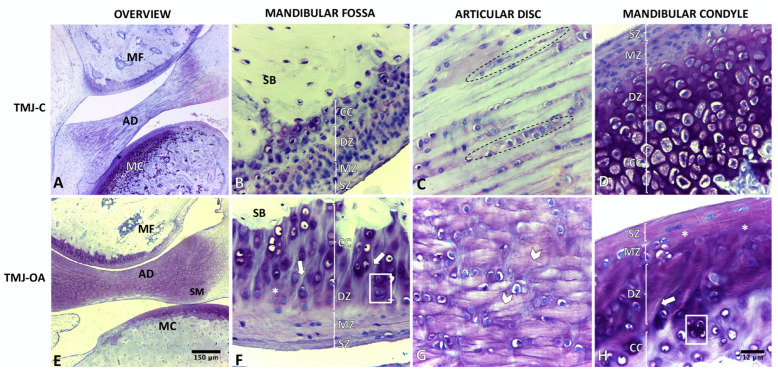
Temporomandibular joint of rabbit (*Oryctolagus cuniculus*). TMJ-C: healthy temporomandibular joint (**A**–**D**); TMJ-OA: temporomandibular joint with osteoarthritis (**E**–**H**); MC: mandibular condyle; AD: articular disc; MF: mandibular fossa; SZ: superficial zone; MZ: middle zone; DZ: deep zone; CC: calcified cartilage; SB: subchondral bone; dotted line area: chondrocytes arranged in clusters parallel to the collagen fibers; grid area: chondron clustering near deep fibrillation; asterisk: heterogeneous matrix texture; arrowhead: disorganized collagen fibers; arrow: deep fibrillation. Toluidine blue stain.

**Figure 2 ijms-24-14130-f002:**
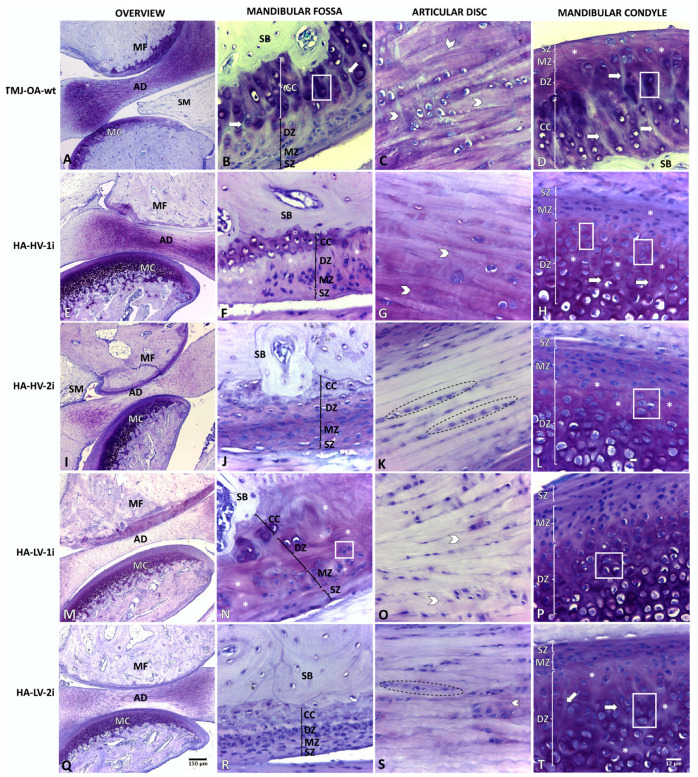
Temporomandibular joint of rabbit (*Oryctolagus cuniculus*). TMJ-OA-wt: osteoarthritic temporomandibular joint without treatment and evaluated at 30 days post-induction (**A**–**D**); HA-HV-1i: TMJ-OA treated with one dose of high-viscosity hyaluronic acid and evaluated 30 days after the treatment (**E**–**H**); HA-HV-2i: TMJ-OA treated with two doses of hyaluronic acid with high viscosity and evaluated 30 days after treatment (**I**–**L**); HA-LV-1i: TMJ-OA treated with one dose of low-viscosity hyaluronic acid and evaluated 30 days after treatment (**M**–**P**); HA-LV-2i: TMJ-OA joint treated with two doses of low-viscosity hyaluronic acid and evaluated 30 days after treatment (**Q**–**T**); MC: mandibular condyle; AD: articular disc; MF:, mandibular fossa; SZ: superficial zone; MZ: middle zone; DZ: deep zone; CC: calcified cartilage; SB: subchondral bone; dotted line area: chondrocytes arranged in clusters parallel to the collagen fibers; grid area: chondron clustering near deep fibrillations; asterisk: heterogeneous matrix texture; arrowhead: disorganized collagen fibers; arrow: deep fibrillations. Toluidine blue stain.

**Figure 3 ijms-24-14130-f003:**
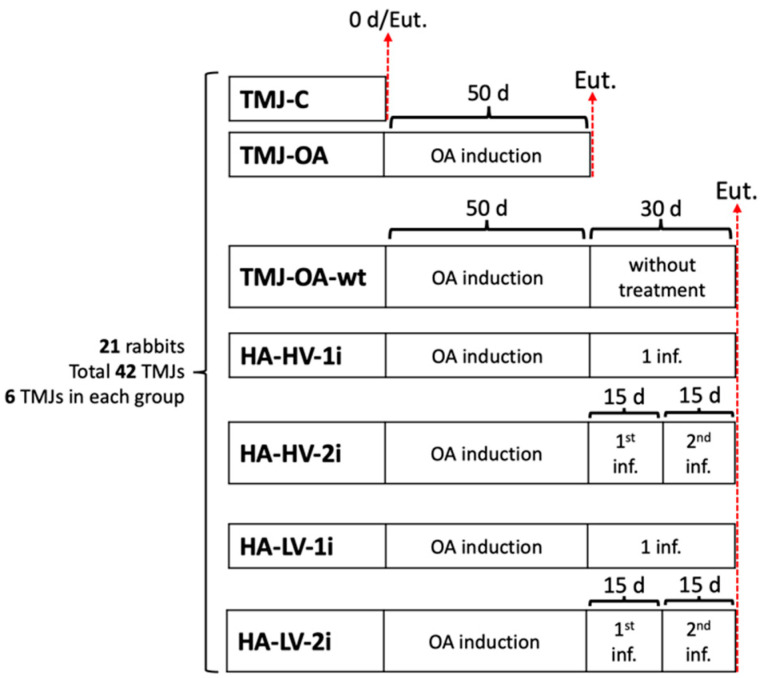
Diagram of experimental groups.

**Table 1 ijms-24-14130-t001:** Score according to the OARSI scale.

Group	MC Grade Median (IR)	MC Stage Median (IR)	MF Grade Median (IR)	MF Stage Median (IR)
TMJ-OA	3.5 ^a^ (3.3–3.5)	4 (4–4)	3.1 ^h^ (3–3.4)	4 (4–4)
TMJ-OA-wt	4 ^b,c,d,e^ (4–4.5)	4 (4–4)	4 ^g,i,j,k^ (4–4.5)	4 (4–4)
HA-HV-1i	3 ^b^ (2.8–3)	4 (3–4)	3 ^g,l^ (2.8–3)	4 (4–4)
HA-HV-2i	2.3 ^a,c,f^ (2.3–3)	4 (3–4)	2 ^h,i,l,m^ (2–2.5)	4 (3–4)
HA-LV-1i	2.8 ^d^ (2.3–3)	4 (3.5–4)	3 ^j,m^ (3–3)	4 (4–4)
HA-LV-2i	3 ^e,f^ (3–3.3)	4 (4–4)	2.8 ^k^ (2–2.8)	4 (4–4)

MC: mandibular condyle; MF: mandibular fossa; IR: interquartile range ^a^: *p* = 0.006; ^b^: *p* = 0.01; ^c^: *p* = 0.0001; ^d^: *p* = 0.002; ^e^: *p* = 0.02; ^f^: *p* = 0.04; ^g^: *p* = 0.02; ^h^: *p* = 0.004; ^i^: *p* = 0.0001; ^j^: *p* = 0.03; ^k^: *p* = 0.003; ^l^: *p* = 0.04; ^m^: *p* = 0.02.

**Table 2 ijms-24-14130-t002:** Histological characteristics of TMJ-OA in the different treatment groups.

Structure		HA-HV-1i	HA-HV-2i	HA-LV 1i	HA-LV 2i
Mandibular condyle	SZ	Slightly irregular surface. It presents small and flat or round chondrocytes, which are aligned parallel to the collagen fibers and to the surface.			
		Abrasion is noted in some areas *.			Abrasion is noted in some areas *.
	MZ	Proliferation of chondrocytes is observed, which are arranged in isolation. The matrix is reactive with increased cationic staining with Toluidine blue.			
		Rarefaction of the matrix is observed. In some areas, there are traces of fibrillation.		Rarefaction of the matrix is observed. In some areas, there are traces of fibrillation.	Rarefaction of the matrix is observed. In some areas, there are traces of fibrillation.
	DZ	Traces of deep fibrillations and focal rarefaction with clustered chondrocytes and increased staining around the chondrons are observed.			
			Traces of deep fibrillations are less evident.		Decreased density of chondrocytes. In some areas, chondrons are observed close to the traces of deep fibrillations.
Articular disc	CZ	Collagen fibers arranged in parallel, with chondrocytes aligned to them. Chondrocytes are found within the cartilage matrix.			
	PZ	Hypertrophic chondrocytes are randomly arranged among disorganized collagen fibers.			
Mandibular fossa	SZ	Slightly irregular surface. Collagen fibers parallel to the surface with scant cellularity. The limits between the SZ and MZ are not very evident.			
		Abrasion is noted in some areas.		Abrasion is noted in some areas.	Abrasion is noted in some areas.
	MZ	Decreased chondrocyte density. Nuclear anisocytosis.			
		Heterogeneous matrix, with edema and focal rarefaction.		Heterogeneous matrix, with edema and focal rarefaction. Increased collagen formation.	
	DZ	Decreased chondrocyte density. Nuclear anisocytosis.			
		Heterogeneous matrix, with edema and focal rarefaction.		Heterogeneous matrix, with edema and focal rarefaction. Increased collagen formation.	

HA-HV-1i: TMJ-OA treated with one dose of high-viscosity hyaluronic acid and evaluated 30 days after treatment; HA-HV-2i: TMJ-OA treated with two doses of high-viscosity hyaluronic acid and evaluated 30 days after treatment; HA-LV-1i: TMJ-OA treated with one dose of low-viscosity hyaluronic acid and evaluated 30 days after treatment; HA-LV-2i: TMJ-OA treated with two doses of low-viscosity hyaluronic acid and evaluated 30 days after the treatment; SZ: superficial zone; MZ: middle zone; DZ: deep zone; CZ: central zone; PZ: peripheral zone; * Abrasion: focal loss of the superficial portion of superficial zone.

## Data Availability

The data presented in this study are available at https://doi.org/10.6084/m9.figshare.24107085.v1 (accessed on 12 September 2023).

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
