# Peer review of "Comparison of Four Treatment Protocols with Intra-Articular Medium Molecular Weight Hyaluronic Acid in Induced Temporomandibular Osteoarthritis: An Experimental Study"

_ijms, 2023, doi:10.3390/ijms241814130_

Round 1

Reviewer 1 Report

In this study the Authors aimed to compare the effect between the single intra-articular infiltration  and two infiltrations of medium molecular weight hyaluronic acid of high viscosity and low viscosity on the histopathological characteristics of the temporomandibular joint osteoarthritis (OA) induced in rabbit.

My specific comments are as follows:

a) Table 1 shows the results obtained from the assessments according to the OARSI scale. The data are described only by reporting the mean value of the histopathologic scores. This data should be completed by also showing a measure describing the variability of the dataset (e.g., standard deviation)

b) Given the small sample size and because non-continuous variables were evaluated, the Authors should consider expressing the data as median and interquartile range.

In addition, since the statistical analysis included multiple groups being compared with each other, in my opinion the Authors should consider applying nonparametric tests for multiple comparisons (e.g. Kruskal-Wallis) followed by post-hoc analysis

Authors should check and verify that the statistical analysis was performed using the appropriate tests

c) The legend of histologic figures should also report the corresponding magnification for each image

d) In the discussion and conclusions the Authors should more strongly highlight the novel key results of the present study that might introduce new aspects and new potential advances from a clinical point of view

Minor point:

Diagram of experimental group was identified as figure1, but it is mentioned in the text after figure 2 and figure 3. The Author should check and modify the numbering of the figures according to the order of citation in the text

Reviewer 2 Report

Manuscript review: Comparison of four treatment protocols with intra-articular medium molecular weight hyaluronic acid in induced temporomandibular osteoarthritis.

Title and abstract:

- It is not obligatory, but for ease of reading, I recommend adding information about the type of research at the end of the title, e.g. experimental, animal.

Introduction:

- It is worth noting that hyaluronic acid is the main component of natural synovial fluid, and its injection is more like supplementation than administration of a drug.

- Consider listing (1) Background, (2) Rationale, and (3) Aim as separate subsections.

- There are too few references to the latest papers.

Results:

- I have no comments for this section.

Discussion:

- The Limitations description is present, but for clarity it would be better if it was a separate subsection.

- There are too few references to the latest papers.

Methods:

- I have no comments for this section.

Back Matter:

- If the protocol of this study has been previously published, please provide a reference. If not, please provide that information.

- There is no statement about data availability.

Reference:

- The topic of using hyaluronic acid in the treatment of TMJs osteoarthritis is widely discussed in the literature. Supplement references with the latest papers, preferably with a high level of evidence. Keywords "hyaluronic temporomandibular" with filters "Systematic review" and "last 5 years" give in PubMed several items that should be worth your attention.

Round 2

Reviewer 1 Report

No further modification are needed